# Platelet-Rich Plasma: New Performance Understandings and Therapeutic Considerations in 2020

**DOI:** 10.3390/ijms21207794

**Published:** 2020-10-21

**Authors:** Peter Everts, Kentaro Onishi, Prathap Jayaram, José Fábio Lana, Kenneth Mautner

**Affiliations:** 1Gulf Coast Biologics, Research and Science Division, Fort Myers, FL 33916, USA; 2Department of PM&R and Orthopedic Surgery, University of Pittsburg Medical Center, Pittsburgh, PA 15213, USA; kenonishi918@gmail.com; 3Department of Physical Medicine & Rehabilitation, Baylor College of Medicine, Houston, TX 77030, USA; prathap.jayaram@gmail.com; 4The Bone and Cartilage Institute, Indaiatuba, Sao Paulo, Brazil; josefabiolana@gmail.com; 5Emory Sports Medicine and Primary Care Sports Medicine, Emory University, Atlanta, GA 30329, USA; kmautne@emory.edu

**Keywords:** platelet-rich plasma, regenerative medicine, platelet dosing, neutrophils, monocytes, lymphocytes, inflammation, angiogenesis, serotonin, analgesic effects, immunomodulation, rehabilitation

## Abstract

Emerging autologous cellular therapies that utilize platelet-rich plasma (PRP) applications have the potential to play adjunctive roles in a variety of regenerative medicine treatment plans. There is a global unmet need for tissue repair strategies to treat musculoskeletal (MSK) and spinal disorders, osteoarthritis (OA), and patients with chronic complex and recalcitrant wounds. PRP therapy is based on the fact that platelet growth factors (PGFs) support the three phases of wound healing and repair cascade (inflammation, proliferation, remodeling). Many different PRP formulations have been evaluated, originating from human, in vitro, and animal studies. However, recommendations from in vitro and animal research often lead to different clinical outcomes because it is difficult to translate non-clinical study outcomes and methodology recommendations to human clinical treatment protocols. In recent years, progress has been made in understanding PRP technology and the concepts for bioformulation, and new research directives and new indications have been suggested. In this review, we will discuss recent developments regarding PRP preparation and composition regarding platelet dosing, leukocyte activities concerning innate and adaptive immunomodulation, serotonin (5-HT) effects, and pain killing. Furthermore, we discuss PRP mechanisms related to inflammation and angiogenesis in tissue repair and regenerative processes. Lastly, we will review the effect of certain drugs on PRP activity, and the combination of PRP and rehabilitation protocols.

## 1. Introduction

Autologous platelet-rich plasma (PRP) is the processed liquid fraction of autologous peripheral blood with a platelet concentration above the baseline [1]. PRP therapies have been used for various indications for more than 30 years, resulting in considerable interest in the potential of autologous PRP in regenerative medicine. The term orthobiologics has recently been introduced for the treatment of musculoskeletal (MSK) disorders, with promising results for the regenerative capacity of the heterogeneous biological active PRP cellular cocktail. Currently, PRP therapies are suitable treatment options with clinical benefits, with encouraging patient outcomes reported [2,3,4]. However, inconsistencies in patient outcomes and new insights have challenged the practicality of PRP clinical applications [5,6,7]. One reason might be the large number and variability of commercially available PRP and PRP-like systems. These devices vary in PRP collection volumes, preparation protocols that result in distinctive PRP properties and bioformulations. Furthermore, the lack of consensus on standardization of PRP preparation protocols, with adequate reporting on bioformulations in clinical applications, contributes to inconsistencies in reported outcomes. Several attempts have been made to characterize and classify PRP or blood-derived products in regenerative medicine applications. In addition, platelet-derivates, like human platelet lysate, have been proposed in orthopedics and in-vitro stem cell research [8].

One of the first reviews on PRP was published in 2006 [9]. The main focus of the review was on platelet function and mode of action, the effect of PRP on the various stages of the healing cascade, and the central role for platelet-derived growth factors in various PRP indications. In the early years of PRP research, the main interest in PRP or PRP-gel was the presence of several platelet growth factors (PGFs) and their specific functions, as shown in Table 1. In this article, we will extensively discuss recent developments in the different PRP granular structures and platelet cell membrane receptors and their effects on the immunomodulatory actions of the innate and adaptive immune system. In addition, the roles of individual cells that can be present in PRP treatment vials will be discussed along with detailed information on their effects on tissue regenerative processes. Furthermore, recent advances in understanding PRP bioformulations, platelet dosing, the specific roles of particular leukocytes, and the effects of PGF concentrations and cytokines on mesenchymal stem cell (MSC) trophic effects [10] will be described, including the pivotal roles of PRP in targeting different cells and tissue environments following cell-signaling and paracrine effects [11]. Likewise, we will discuss PRP mechanisms related to inflammation and angiogenesis in tissue repair and regenerative processes. Lastly, we will review the analgesic effects of PRP [12], the effect of certain drugs on PRP activity, and the combination of PRP and rehabilitation protocols.

## 2. The Rationale for Clinical Platelet-Rich Plasma Therapies

PRP preparations have gained increasing popularity with widespread use in diverse medical fields. The underlying scientific rationale for PRP therapy is that an injection of concentrated platelets at sites of injury may initiate tissue repair via the release of many biologically active factors (growth factors, cytokines, lysosomes) and adhesion proteins that are responsible for initiating the hemostatic cascade, synthesis of new connective tissue, and revascularization. Additionally, plasma proteins (e.g., fibrinogen, prothrombin, and fibronectin) are present in the platelet-poor plasma fraction (PPP). PRP concentrates can stimulate the supraphysiological release of growth factors to jump-start healing in chronic injuries and accelerate the acute injury repair process [9]. At all stages of the tissue repair process, a wide variety of growth factors, cytokines, and locally acting regulators contribute to most basic cell functions via endocrine, paracrine, autocrine, and intracrine mechanisms. The main advantages of PRP include its safety and the ingenious preparation techniques of current commercial devices to prepare a biologic that can be used in a broad application profile [14]. Most importantly, PRP is an autologous product with no known adverse effects, in contrast to the commonly used corticosteroids [15]. However, there are no clear regulations regarding the formulation and composition of an injectable PRP composition [16], and PRP compositions vary greatly in platelet, white blood cell (WBC) content, red blood cell (RBC) contamination, and PGF concentrations [17,18].

## 3. PRP Terminology and Classification

The development of PRP products to stimulate tissue repair and regeneration has been an important research field in biomaterial and pharmaceutical sciences for decades. The tissue healing cascade incorporates many players, including platelets with their growth factor and cytokine granules, leukocytes, fibrin matrix, and many other cytokines, which work synergistically. During this cascade, a complex coagulation process occurs, consisting of platelet activation and subsequent release of the contents of dense and α-platelet granules, polymerization of fibrinogen (released by platelets or free in the plasma) into a fibrin mesh, and platelet plug development [19].

### 3.1. “Generic” PRP to Mimic the Onset of Healing

Originally, the term “platelet-rich plasma (PRP)” was referenced as a platelet concentrate used in blood transfusion medicine, where it still is used today. At first, these PRP products were only used as fibrin tissue adhesives, and platelets were only used to support a stronger fibrin polymerization to improve tissue sealing but not as healing stimulators [20]. After that, PRP technologies were designed to mimic the initiation of the healing cascade. Subsequently, PRP technology has been summarized by its ability to introduce and release growth factors to a local microenvironment. This enthusiasm for PGF delivery often overshadowed the vital roles of other constituents present in these blood-derived products, which was further aggravated by a lack of scientific data, mystical belief, commercial interests, and lack of standardization and classification [21].

The biology of PRP concentrates is as complex as blood itself and likely more complex than traditional pharmaceutical drugs. PRP products are living biomaterials, and the outcomes of clinical PRP applications are dependent on the intrinsic, versatile, and adaptive characteristics of the patient’s blood, including various other cell constituents that may be present in the PRP specimen [17] and the interaction with the recipient local microenvironment, which can be in an acute or chronic state.

### 3.2. Confusing PRP Terminology and Synopsis of Proposed Classification Systems

Over the years, practitioners, scientists, and companies have suffered from the initial misperceptions and shortcomings regarding PRP products and their different terminologies. Some authors have defined PRP as only platelets, whereas others note that PRP also contains increased concentrations of RBCs, various leukocytes, fibrin, and bioactive proteins. Consequently, many different PRP bioformulations have been introduced into clinical practice. Disappointingly, the literature often lacks detailed bioformulation descriptions [22]. Failures in product preparation standardization and subsequent development of a classification system led to the use of a magnitude of PRP-like products that are described by different terminologies and abbreviations (Table 2). It is not surprising that variations in PRP preparations resulted in inconsistent patient outcomes.

Kingsley was the first to use the term “platelet-rich plasma” back in 1954 [23]. Many years later, Ehrenfest et al. [24] proposed the first classification system based on three main variables (i.e., platelets, leukocytes, and fibrin content), which divided many PRP products into four main categories: P-PRP, LR-PRP, pure platelet-rich fibrin (P-PRF), and leucocyte-rich PRF (L-PRF). These products were prepared by either a fully automated-closed system or manual protocol. In parallel, Everts et al. [22] emphasized the importance of mentioning the presence of leukocytes in the PRP preparations. They also suggested using proper terminology that indicated non-activated or activated versions of the PRP preparation and platelet gel.

Delong et al. [25] proposed the PRP classification system called platelet, activation, white blood cells (PAW) based on the absolute number of platelets, including four platelet concentration ranges. Other parameters included the use of platelet activators and the presence or absence of leukocytes (i.e., neutrophils). Mishra et al. [26] suggested a similar classification system. A few years later, Mautner and associates described a more refined and detailed classification system (PLRA) [27]. The authors demonstrated that it is important to describe the absolute platelet count, leukocyte content (positive or negative), the percentage of neutrophils, RBC (positive or negative), and whether exogenous activation was used. In 2016, Magalon et al. [28] published the DEPA classification based on the dose of injected platelets, production efficiency, the PRP purity obtained, and the activation process. Subsequently, Lana and co-workers introduced the MARSPILL classification system with a significant focus on peripheral blood mononuclear cells [29]. Most recently, the Scientific Standardization Committee has advocated for the adoption of the classification system of the International Society on Thrombosis and Hemostasis, which is based on a series of consensus recommendations for standardizing the use of platelet products for regenerative medicine applications, including frozen and thawed platelet products [30].

Based on the many unsuccessful attempts by various practitioners and researchers to present a PRP classification system to standardize PRP production, definitions, and formulations that would be adopted by clinicians, it is fair to conclude that this most likely will not happen in the years to come. Furthermore, clinical PRP product technologies continue to develop, and scientific data indicate the need for different PRP formulations to treat different pathologies under specific conditions. Thus, we expect the parameters and variables for ideal PRP production will continue to grow in the future (Table 3).

### 3.3. PRP Preparation Methods Are a Work in Progress

Based on the PRP terminologies and product descriptions, several classification systems have been published for different PRP formulations [31]. Regrettably, there is no consensus on a comprehensive classification system for PRP or any other autologous blood and blood-derived preparations. Ideally, a classification system should focus on the various PRP characteristics, definitions, and appropriate nomenclature that are relevant for therapeutic decision-making to treat patient-specific conditions. Currently, ortho-biological applications classify PRP into three groups: pure platelet-rich fibrin (P-PRF), leukocyte-rich PRP (LR-PRP), and leukocyte-poor PRP (LP-PRP) [32]. Although more specific then a generic PRP product definition, the LR-PRP and LP-PRP categories are significantly lacking any specificity regarding the leukocyte content. Leukocytes greatly impact the intrinsic biology of chronic tissue lesions due to their immune and host-defense mechanisms. Therefore, PRP biological preparations containing specific leukocytes can significantly contribute to immune modulation and tissue repair and regeneration. More specifically, lymphocytes are abundantly present in PRP, producing insulin-like growth factors and supporting tissue remodeling.

Monocytes and macrophages play key roles in immunomodulatory processes and tissue repair mechanisms. The importance of neutrophils in PRP is unclear. Systematic reviews identified LP-PRP as the preferred PRP formulation to achieve effective treatment outcomes for joint OA [33]. However, Lana et al. [34] opposed the use of LP-PRP in knee OA treatment, suggesting that particular leukocytes play an important role in the inflammatory process preceding tissue regeneration due to their release of both pro and anti-inflammatory molecules. They found that the combination of neutrophils and activated platelets could have a more positive than detrimental effect on tissue repair. They also indicated that the plasticity of monocytes is important for the non-inflammatory and reparative roles in tissue repair.

The reporting of PRP preparation protocols in clinical studies has been highly inconsistent, and the majority of published studies do not present the PRP preparation methods needed for protocol reproducibility. A clear consensus across treatment indications is non-existent, making it difficult to compare PRP products and their related therapy outcomes. In the majority of reported cases, platelet concentrate therapies are all grouped under the term “PRP,” even for the same clinical indication [21]. For some medical fields (e.g., OA and tendinopathies), progress has been made in understanding the variations in the PRP formulations, delivery routes, platelet function, and other PRP constituents influencing tissue repair and tissue regeneration. However, further research is needed to develop a consensus regarding PRP terminologies related to the PRP bioformulations to adequately and safely treat certain pathologies and conditions.

### 3.4. The Current Status of PRP Classification Systems

The employment of autologous PRP biological therapies is troubled by the heterogenicity in PRP formulations, inconsistencies in nomenclature, and poor standardization of evidence-based guidelines (i.e., there are multitudes of preparation methods generating a clinical treatment vial). Predictably, the absolute PRP content, purity, and biological properties of PRP and related products vary widely and impact the biological efficacy and clinical trial outcomes. The choice of PRP preparation device introduces the first critical variable. In clinical regenerative medicine, practitioners can use two distinctly different PRP preparation devices and methods. One preparation uses standard blood cell separators, which operate on a full unit of autologously harvested blood. This approach uses continuous-flow centrifuge bowl or disk separation technology coupled with hard and soft centrifugation steps. These devices are mostly used intra-operatively. An alternative approach uses gravitational centrifugation techniques and devices. High *G*-force centrifugation is used to isolate the buffy coat layer from a unit of blood, which contains platelets and leukocytes. These concentration devices are smaller than the blood cell separators and used at the point-of-care. Differences in the *G*-force and centrifugation time result in significant differences in yields, concentration, purity, viability, and activation status of the isolated platelets. Many types of commercial PRP preparation devices are available in the latter category, resulting in further variations in product content [35].

The lack of a consensus in PRP preparation methods and validation continues to contribute to inconsistencies in PRP therapies, with enormous differences in PRP formulation, specimen quality, and, thus, clinical outcomes. Existing commercially available PRP devices have been validated and registered according to proprietary manufacturers’ specifications, addressing different variables among the presently obtainable PRP devices.

## 4. Understanding In Vitro and In Vivo Platelet Dosing

The therapeutic actions of PRP and other platelet concentrates stem from the release of a multitude of factors involved in tissue repair and regeneration. Following platelet activation, a platelet plug is formed, which acts as a temporary extracellular matrix, allowing cells to proliferate and differentiate [1]. Therefore, it is fair to assume that higher platelet dosages will generate an elevated local concentration of released platelet bioactive factors. However, the correlation between platelet dose, concentration, and the concentration of released platelet bioactive growth factors and agents may not be controlled because there are marked differences in baseline platelet counts between individual patients [36], and differences exist between PRP preparation methods [37,38]. Likewise, several platelet growth factors involved in tissue repair mechanisms reside in the plasma fraction of PRP (e.g., hepatic growth factor and insulin-like growth factor 1). Therefore, higher platelet dosing does not affect the repair potential of these growth factors.

In vitro PRP research is popular because the different parameters in these studies can be precisely controlled, and study results obtained quickly. Several studies have demonstrated that cells respond to PRP in a dose-dependent manner. Nguyen and Pham [39] showed that very high concentrations of GF are not necessarily advantageous for cell stimulatory processes and may be counterproductive. Some in vitro studies have indicated that high PGF concentrations may have detrimental effects [40]. One reason could be that the quantity of cell membrane receptors is limited. Thus, once the PGF levels are too high compared to the available receptors, they negatively affect cell function.

### The Significance of In Vitro Data on Platelet Concentrations

Although in vitro studies have many advantages, they also have some weaknesses. In vitro, due to tissue architecture and cell organization, there is a continuous interplay between many different cell types within any tissue, making it difficult to replicate in vitro in a two-dimensional monoculture setting. Cell density, which can affect cell signaling pathways, is usually less than 1% of the tissue situation. The two-dimensional in vitro culture dish organization precludes cells from being exposed to an extracellular matrix (ECM). Furthermore, typical culturing techniques lead to the accumulation of cellular waste products and continuous nutrient consumption. Thus, in vitro culturing does not resemble any homeostatic conditions, the tissue oxygen supply, or the sudden exchange of media, making it difficult to translate in vitro PRP dosing results to clinical practice.

Conflicting results have been published comparing the clinical effects of PRP to in vitro studies for specific cells, tissue types, and platelet concentrations. Graziani et al. [41] found that in vitro, the maximum effect on the proliferation of osteoblasts and fibroblasts was achieved at a PRP platelet concentration that was 2.5-fold times higher than the baseline value. In contrast, clinical data presented by Park and associates [42] indicated that more than a 5-fold increase in PRP platelet levels above baseline was required to induce a positive outcome after spinal fusion. Similar contradictory outcomes have been reported between in vitro tendon proliferation data and clinical outcome studies [43,44].

## 5. A Contemporary PRP Formulation: “Clinical PRP”

PRP treatment protocols have evolved immensely over the past 10 years. Through experimental and clinical research, we now have a better understanding of platelet and other cellular physiology. Furthermore, several high-quality systematic reviews, meta-analyses, and randomized controlled trials signify the effectiveness of PRP biotechnology in many medical fields, including dermatology [45], cardiac surgery [46], plastic surgery [47], orthopedic surgery [48], pain management [49], spinal disorders [50], and sports medicine [51,52].

PRP is currently characterized by its absolute platelet concentration, thereby shifting from the initial definition of PRP consisting of a platelet concentration above baseline values [1] to a minimum platelet concentration of more than 1 × 10^6^/µL or an approximately five-fold increase in platelets from baseline [53]. In the extensive review by Fadadu et al. [35], 33 PRP systems and protocols were evaluated. Some of these systems produced final PRP preparations with a platelet count less than that of whole blood. They reported a PRP platelet factor increase as low as 0.52 with a single spin kit (Selphyl^®^) [54]. In contrast, the dual-spin EmCyte Genesis PurePRPII^®^ device produced the highest platelet concentration (1.6 × 10^6^/µL) [35].

It is apparent that in vitro and animal methodologies are not ideal study settings for successful translation into clinical practice. Likewise, device comparison studies do not support decision making, as they indicate a large variation in platelet concentrations among PRP devices [21]. Fortunately, an increase in the understanding of the cellular functions in PRP that affect treatment outcomes was made possible by proteomics-based techniques [55] and profiling [56]. Until there is consensus on standardized PRP preparations and formulations, PRP should follow a clinical PRP recipe to contribute to substantial tissue repair mechanisms and progressive clinical outcomes.

### 5.1. Clinical PRP Recipe

Presently, effective clinical PRP (C-PRP) has been characterized as a complex composition of autologous multicellular components in a small volume of plasma that is acquired from a fraction of peripheral blood after centrifugation. After centrifugation, according to the different cellular densities (where platelets have the lowest density), PRP and their non-platelet cellular constituents can be retrieved from the concentration device, as demonstrated in Figure 1.

C-PRP meets the prerequisites (i.e., tissue type-dependent, platelet dosage, minimal RBC contamination, addition or removal of particular leukocytes) to produce significant clinical outcomes. These C-PRP qualifications, combined with elucidating the activities of different PDGFs, platelet proteins, cytokines, and chemokines, contribute to the understanding of the fundamental tissue repair mechanisms involving mitogenesis, angiogenesis, chemotaxis, and extracellular matrix formation.

#### 5.1.1. Platelet Granules

In early clinical PRP applications, α-granules were the most cited intra-platelet structures because of the presence of coagulation factors, a large number of PDGFs, and regulators of angiogenesis but minimal thrombotic functions. Additional factors included less famous chemokine and cytokine constituents, such as platelet factor 4 (PF4), pro-platelet basic protein, P-selectin (activator of integrin), and the chemokine RANTES (Regulated upon Activation, Normal T Cell Expressed and Presumably Secreted). The overall functions of these specific platelet granule constituents are to recruit and activate other immune cells or induce endothelial cell inflammation (Figure 2) [57].

The dense granule constituents like ADP, serotonin, polyphosphates, histamine, and epinephrine are more implicit as modifiers of platelet activation and thrombus formation. Most importantly, many of these elements have immune cell-modifying effects. Platelet ADP is recognized by the P2Y12ADP receptor on dendritic cells (DCs), increasing antigen endocytosis. DCs (antigen-presenting cells) are critical for initiating T-cell immune responses and govern the protective immune response [58], linking the innate and adaptive immune system. Moreover, platelet adenosine triphosphate (ATP) signals through the T-cell receptor P2X7, which results in an increase in the differentiation of CD4 T-helper cells to pro-inflammatory T helper 17 (Th17) cells [59]. Other platelet dense granule constituents (e.g., glutamate and serotonin) induce T-cell migration and increase monocyte differentiation into DCs, respectively [60]. In PRP, these dense granule–derived immune modifiers are highly enriched and have substantial immune functions.

The number of potential interactions, both direct and indirect, between platelets and other (receptor) cells is wide-ranging. As a result, numerous inflammatory effects can be induced by PRP when applied in a local, pathological, tissue environment.

#### 5.1.2. Platelet Concentration

C-PRP should contain a clinical dose of concentrated platelets to produce beneficial therapeutic effects. The platelets in C-PRP should stimulate cell proliferation, synthesis of mesenchymal and neurotrophic factors, contribute to chemotactic cell migration, and stimulate immunomodulatory activities, illustrated in Figure 3 [62,63].

Marx was the first to demonstrate the enhancement of bone and soft tissue healing with a minimum platelet count of 1 × 10^6^/µL [1]. These results were confirmed in a transforaminal lumbar fusion study that demonstrated significantly more fusion when the platelet dose was greater than 1.3 × 10^6^ platelets /µL [64]. Moreover, Giusti et al. [65] revealed that a dose of 1.5 × 10^9^ platelets/mL is needed for tissue repair mechanisms to induce a functional angiogenic response through endothelial cell activity. In this latter study, higher concentrations reduced the angiogenic potential of platelets in follicular and perifollicular angiogenesis. Furthermore, earlier data indicate that the PRP dose also affects the magnitude of the therapy outcome [66]. Therefore, to significantly induce an angiogenic response and stimulate cell proliferation and cell migration, C-PRP should contain at least 7.5 × 10^9^ deliverable platelets in a 5-mL PRP treatment vial.

Apart from dose-dependency, the effects of PRP on cell activity appear to be highly time-dependent. Soffer et al. [73] indicated that short-term exposure to human platelet lysate stimulates bone cell proliferation and chemotaxis. In contrast, long-term PRP exposure results in decreased levels of alkaline phosphatase and mineral formation.

#### 5.1.3. Deleterious Red Blood Cells

RBCs are responsible for transporting oxygen to tissues and removing carbon dioxide from tissues to the lungs [74]. They have no nucleus and are made of protein-bound heme molecules. Iron and heme components inside RBCs facilitate the binding of oxygen and carbon dioxide. Normally, the RBC life cycle is approximately 120 days. They are removed from circulation by macrophages by a process termed RBC senescence. Under conditions of shear forces (e.g., whole blood phlebotomy procedures, immune-mediated processes, oxidative stress, or inadequate PRP concentration protocols), RBCs in the PRP specimens could become damaged. As a consequence, the RBC cell membrane disintegrates and releases toxic hemoglobin (Hb), measured as plasma-free hemoglobin (PFH), hemin, and iron [75]. PFH and its degradation products (heme and iron) collectively lead to detrimental and cytotoxic effects to tissues, causing oxidative stress, loss of nitric oxide, activation of inflammatory pathways, and immunosuppression. These effects ultimately lead to microcirculatory dysfunction, local vasoconstriction with vascular damage, and significant tissue injury.

Most importantly, when C-PRP containing RBCs is delivered to tissues, it causes a local response called eryptosis, which triggers the release of a potent cytokine, macrophage migration inhibitory factor [76]. This cytokine inhibits the migration of monocytes and macrophages. It exerts profound pro-inflammatory signals to surrounding tissues that inhibit the migration of stem cells and fibroblast proliferation and causes significant local cellular dysfunction. Therefore, limiting RBC contamination in PRP preparations is important. Moreover, the role of RBCs in tissue regeneration has never been established. Adequate C-PRP centrifugation and preparation processes typically reduce or even eliminate the presence of RBCs, thereby avoiding the detrimental consequences of hemolysis and eryptosis.

#### 5.1.4. Leukocytes in C-PRP

The presence of leukocytes in PRP preparations are processing device- and preparation protocol-dependent. In plasma-based PRP devices, leukocytes are completely eliminated; however, in buffy coat PRP preparations, leukocytes are significantly concentrated [77]. Leukocytes greatly influence the intrinsic biology of acute and chronic tissue conditions because of their immune and host-defense mechanisms. These characteristics will be discussed further below. Therefore, the presence of specific leukocytes in C-PRP can cause significant cellular and tissue effects. More specifically, different PRP buffy coat systems utilize different preparation protocols, thereby producing different neutrophil, lymphocyte, and monocyte cell ratios in PRP [78]. Eosinophils and basophils are not measurable in PRP formulations as their cell membrane is too fragile to withstand the centrifugal processing forces.

##### Neutrophils

Neutrophils are essential leukocytes in numerous healing pathways that create dense barriers against invading pathogens [79] in conjunction with anti-microbial proteins present in platelets [80]. The presence of neutrophils is determined based on the C-PRP treatment objectives. Exacerbated tissue inflammatory levels can be necessary in chronic wound care PRP biological treatments [6], or applications directed towards bone growth or healing [81]. Importantly, additional neutrophil functions have been uncovered in several models, emphasizing their roles in angiogenesis and tissue restoration [82]. However, neutrophils can also cause harmful effects and, thus, are not indicated for some applications. Zhou and Wang demonstrated that the use of PRP rich in neutrophils could result in a higher collagen type III to collagen type I ratio, adding to fibrosis and decreased tendon strength [83]. Other neutrophil-mediated deleterious properties are the release of inflammatory cytokines and matrix metalloproteinases (MMPs) that promote pro-inflammatory and catabolic effects when applied to tissues [84].

##### Lymphocytes

In C-PRP, mononuclear T and B lymphocytes are more concentrated than any other leukocytes. They are critically involved in cell-mediated cytotoxic adaptive immunity. Lymphocytes can elicit a cell response to fight infection and adapt to intruders [85]. Furthermore, T lymphocyte-derived cytokines (interferon-γ [IFN-γ] and interleukin-4 [IL-4]) strengthen macrophage polarization [86]. Weirather et al. [87] demonstrated that regular T lymphocytes indirectly contribute to tissue healing in a mouse model by modulating monocyte and macrophage differentiation.

##### Monocytes—Multipotential Repair Cells

Depending on the PRP preparation devices used, monocytes may be prominent in PRP treatment vials or absent. Unfortunately, their manifestation and regenerative capabilities are rarely discussed in the literature. Therefore, little attention is given to monocytes in preparation methods or final formulations. Monocyte populations are heterogeneous and originate from progenitor cells in the bone marrow via hematopoietic stem cell pathways and traffic via the bloodstream to peripheral tissues depending on the microenvironmental stimuli. During homeostasis and inflammation, circulating monocytes leave the bloodstream and are recruited to injured or degenerated tissues. They can act either as effector cells or progenitors of macrophages (MΦs). Monocytes, macrophages, and dendritic cells represent the mononuclear phagocyte system (MPS) [88]. A typical feature of the MPS is the plasticity in their gene expression patterns and functional overlap between these cell types. In degenerated tissues, resident macrophages, local-acting growth factors, pro-inflammatory cytokines, apoptotic or necrotic cells, and microbial products initiate the differentiation of monocytes into MPS cell populations [89]. Hypothetically, when C-PRP containing high yields of monocytes is injected in a diseased local microenvironment, monocytes most likely differentiate into MΦs to provoke major cellular changes.

During the monocyte-to-MΦ transition, particular MΦ phenotypes are produced [90]. In the recent decade, a model has been developed that describes the complex mechanism of MΦ activation as a polarization towards two opposite states: MΦ phenotype 1 (MΦ1, classical activation) and MΦ phenotype 2 (MΦ2, alternative activation) [91]. MΦ1 is characterized by inflammatory cytokine secretion (IFN-γ) and nitric oxide production, resulting in an effective pathogen killing mechanism. The MΦ1 phenotype also produces vascular endothelial growth factor (VEGF) and fibroblast growth factor (FGF). The MΦ2 phenotype consists of anti-inflammatory cells with a high phagocytosis capacity. MΦ2 produces extracellular matrix components, angiogenic and chemotactic factors, and interleukin-10 (IL-10). In addition to pathogen defense, MΦ2 can alleviate the inflammatory response and promote tissue repair. Notably, MΦ2 has been subdivided in vitro into MΦ2a, MΦ2b, and MΦ2, depending on the stimulus [92]. An in vivo translation of these subtypes is difficult, as tissues can contain mixed populations of MΦs. Interestingly, pro-inflammatory MΦ1 can switch to pro-repair MΦ2, based on local environmental signaling and IL-4 levels. From these data, it is reasonable to assume that C-PRP preparations containing a high concentration of monocytes and MΦs are likely to contribute to better tissue repair because of their anti-inflammatory tissue repair and cell signaling capabilities.

### 5.2. Confusing Definitions for Leukocyte Fractions in PRP

The presence of leukocytes in PRP treatment vials is dependent on the PRP preparation device and can vary remarkably. Much has been debated about the presence or absence of leukocytes and their contributions to different sub-PRP products, such as PRGF, P-PRP, LP-PRP, LR-PRP, P-PRF, and L-PRF [34,93]. In a recent review, six randomized controlled trials (evidence level 1) and three prospective comparative studies (evidence level 2) with a total of 1055 patients showed that LR-PRP and LP-PRP had similar safety profiles [94]. The authors concluded that the adverse reactions from PRP might not be directly related to the leukocyte concentration. In another study, LR-PRP did not modify systemic or local levels of the pro-inflammatory interleukins (IL-1β, IL-6, IL-8, and IL-17) in OA knees [95]. Those results support the idea that the in vivo role of leukocytes in the bioactivity of PRP might come from the crosstalk between the platelets and leukocytes. This interaction could promote the biosynthesis of other factors (e.g., lipoxins) that counteract or facilitate the resolution of inflammation [96]. After the initial release of inflammatory molecules (arachidonic acid, leukotrienes, and prostaglandins), lipoxin A4 is released from activated platelets to prevent neutrophil activation [34]. It is in this milieu that switches the MΦ phenotypes, from MΦ1 to MΦ2 [96]. Moreover, there has been accumulating evidence indicating that circulating monocytes can differentiate into a variety of non-phagocytic cell types due to their multipotential nature [96].

The type of PRP can influence MSC cultures. LR-PRP can induce significantly higher bone marrow derived MSC (BMMSC) proliferation than pure PRP or PPP samples, with faster release and better biological activity of PGFs [97]. All these properties favor the inclusion of monocytes in PRP treatment vials and acknowledge their immunomodulatory capacity and differentiation potential.

## 6. Innate and Adaptive Immunomodulatory Capacities of PRP

The most well-known physiological role of platelets is the control of hemorrhage, where they accumulate at tissue injury sites and damaged blood vessels. These events are instigated by the expression of integrins and selectins that stimulate platelet adhesion and aggregation. This process is further aggravated by damaged endothelium, with exposed collagen and other subendothelial matrix proteins prompting profound platelet activation. Under these circumstances, a significant role for the von Willebrand factor (vWF) interacting with glycoproteins (GPs), in particular with GP-Ib, has been demonstrated [98]. Following platelet activation, the platelet α-, dense, lysosomal, and T-granules undergo regulated exocytosis and release their contents into the extracellular environment (Figure 2) [99,100].

### 6.1. Platelet Adhesion Molecules

To better comprehend the role of PRP in inflamed tissues and platelets in immune responses, it is crucial to have an understanding of the different platelet surface receptors (integrins) and junctional adhesion molecules (JAM) and how cellular interactions are initiated in innate and adaptive immunity processes.

Integrins are cell surface adhesion molecules found on various cell types and abundantly expressed on platelets. The integrins include a5b1, a6b1, a2b1 LFA-2, (GPIa/IIa), and aIIbb3 (GPIIb/IIIa) [101]. Normally, they exist in a quiescent, low-affinity state. Upon activation, they switch to a high ligand-binding affinity state. On platelets, integrins have dissimilar functions and are involved in the interaction of platelets with several types of leukocytes, endothelial cells, and the extracellular matrix [71]. In addition, the GP-Ib–V–IX complex is expressed on platelet membranes and is the main receptor for binding to von vWF. This interaction mediates the initial contact of platelets with exposed subendothelial structures [102]. Platelet integrins and GP complexes have been implicated in several inflammatory processes, playing an important role in the formation of platelet–leukocyte complexes. Specifically, integrin aIIbb3 is required for the formation of stable complexes by binding to macrophage-1 antigen (Mac-1) receptors on neutrophils via fibrinogen [72].

Platelets, neutrophils, and vascular endothelial cells express specific cell adhesion molecules, known as selectins [103]. Under inflammatory conditions, platelets express P-selectin and neutrophils L-selectin. Upon platelet activation, P-selectin may bind to its ligand PSGL-1 present on neutrophils and monocytes [104]. In addition, PSGL-1 binding initiates an intracellular signaling cascade, leading to neutrophil activation via neutrophil integrins Mac-1 and lymphocyte function-associated antigen-1 (LFA-1). Activated Mac-1 binds to GPIb or GPIIb/IIIa on platelets via fibrinogen, which in turn stabilizes the neutrophil-platelet cell-cell interaction [105]. Moreover, activated LFA-1 can bind to the platelet intercellular adhesion molecule-2, further stabilizing the neutrophil-platelet complex to promote prolonged attachment to cells [106].

### 6.2. Platelets and Leukocytes Play Pivotal Roles in Innate and Adaptive Immune Responses

The body can identify foreign bodies and injured tissues in acute or chronic conditions to initiate the wound healing cascade and inflammatory pathways. The innate and adaptive immune systems protect the host from infection, with essential roles for leukocytes overlapping between both systems, as displayed in Figure 4. Specifically, monocytes, macrophages, neutrophils, and natural killer cells have pivotal roles in the innate system, whereas lymphocytes and their subsets play similar roles in the adaptive immune system [107].

#### 6.2.1. Innate Immune System

The role of the innate immune system is to nonspecifically identify intruding microbes or tissue fragments and stimulate their clearance. Activation of the innate immune system occurs when certain molecular structures, termed surface-expressed pattern recognition receptors (PRRs), bind to pathogen-associated molecular patterns and damage-associated molecular patterns. There are many classes of PRRs, including Toll-like receptors (TLRs) and RIG-1-like receptors (RLRs) [108]. These receptors can activate the major transcription factor nuclear factor kappa B (NF-κB) and regulate multiple aspects of both the innate and adaptive immune response. Interestingly, platelets also express several immunomodulatory receptor molecules on their surface and in the cytoplasm, such as P-selectin, transmembrane protein CD40 ligand (CD40L), cytokines (e.g., IL-1β, TGF-β), and platelet-specific TLRs [109]. Therefore, platelets can interact with various immune cells.

##### Platelet-Leukocyte Interactions in Innate Immunity

Platelets are among the first cells to detect endothelial injury and microbial pathogens as they gain access or invade the bloodstream or tissues. Platelets aggregate and promote the release of the platelet agonists ADP, thrombin, and vWF, leading to platelet activation and expression of platelet chemokine receptors C, CC, CXC, and CX3C, which results in a rapid accumulation of platelets at the site of infection or injury [110].

The innate immune system is genetically predetermined to detect invaders, such as viruses, bacteria, parasites, and toxins, or tissue trauma and wounds. It is a nonspecific system, as any pathogen will be identified as foreign or non-self and rapidly targeted. The innate immune system depends on a group of proteins and phagocytic cells, which identify well-preserved features of the pathogens and quickly activate the immune response to help destroy the invaders, even if the host has never been previously exposed to a particular pathogen [111].

Neutrophils, monocytes, and dendritic cells are the most common innate immune cells in the blood. Their recruitment is required for an adequate early-phase immune response. Platelet-leukocyte interactions regulate inflammation, wound healing, and tissue repair when PRP is used in regenerative medicine applications. TLR-4 on platelets stimulates platelet-neutrophil interactions [112], which regulate the so-called leukocyte oxidative burst by modulating the release of reactive oxygen species (ROS) and myeloperoxidase (MPO) from neutrophils [113]. Furthermore, the platelet-neutrophil interaction with neutrophil degranulation results in the formation of neutrophil-extracellular traps (NETs). NETs are comprised of the neutrophil nucleus and other neutrophil intracellular contents that trap bacteria and kill them by NETosis. The formation of NETs is an essential killing mechanism for neutrophils [114].

Following platelet activation, monocytes can migrate to diseased and degenerative tissues where they perform adhesion activities while secreting inflammatory molecules that may alter chemotaxis and modify proteolytic properties [115]. Additionally, platelets can modulate the effector functions of monocytes by inducing the activation of monocyte NF-κb [67], a critical mediator of the inflammatory response and the activation and differentiation of immune cells. Platelets further facilitate the endogenous oxidative burst in monocytes to boost the destruction of phagocytosed pathogens, and the release of MPO is mediated by direct platelet-monocyte CD40L-MAC-1 interactions [68]. Intriguingly, when P-selectin activates platelets during acute and chronic inflammatory tissue conditions, the platelet-derived chemokines PF4, RANTES, IL-1β, and CXCL-12 prevent monocytes from undergoing spontaneous apoptosis but promote their differentiation into macrophages [116].

DCs originate from bone marrow hematopoietic precursor cells and constitute a unique cell system that induces primary innate immune responses through phagocytosis. DCs can recognize pathogens and tissue damage signals and then migrate to secondary lymphoid organs where they present antigens and activate various T lymphocytes. DCs are classified into conventional DCs, plasmacytoid DCs, and DCs derived from monocytes (mDCs). However, mDCs appear only when there is an inflammatory condition. It is important to note that platelets recruit DCs through the interactions of immunoglobulin JAM receptors and the MAC-1 integrins of neutrophils and monocytes/macrophages [71]. Therefore, DCs are critical cells that are adept at bridging the innate and adaptive immune systems as they will differentiate after PRP platelet-derived growth factors are released [117].

#### 6.2.2. Adaptive Immune System

Following the identification of microbes or tissue damage by the non-specific innate immune system, the specific adaptive immune system takes over. The adaptive system includes B lymphocytes (B cells), which bind antigens, and regular T lymphocytes (Treg), which coordinate the elimination of the pathogens. T cells can be broadly categorized into helper T cells (Th cells) and cytotoxic T cells (Tc cells, also known as T killer cells) [107]. The Th cells are further divided into Th1, Th2, and Th17 cells, with critical functions in inflammation. The Th cells can secrete pro-inflammatory cytokines (e.g., IFN-γ, TNF-β) and several interleukins (e.g., IL-17). They are particularly effective in protecting against intracellular viral and bacterial infections. Th cells stimulate proliferation and differentiation of cells involved in the immunological response. Tc cells are effector cells that eliminate the targeted intracellular and extracellular microbes and cells [118].

Interestingly, the Th2 cells produce IL-4 and influence MΦ polarization, directing MΦs to the regenerative MΦ2 phenotype, while IFN-γ shifts MΦ toward the inflammatory MΦ1phenotype, depending on the dose and timing of the cytokines. Following IL-4 activation, MΦ2 induce the differentiation of Treg cells to Th2 cells, subsequently producing additional IL-4 (positive feedback loop) [119]. Th cells guide MΦ phenotypes to pro-regenerative phenotypes in response to tissue-derived biologics in an IL-4-dependent manner [120]. This mechanism is based on the evidence that Th cells have a pronounced role in both controlling inflammation and tissue repair.

##### Platelet-Leukocyte Interactions in Adaptive Immunity

The adaptive immune system employs antigen-specific receptors and remembers previous pathogen encounters and destroys these pathogens during subsequent encounters with the host. However, these adaptive immune responses are slow to develop. Cognasse et al. [109] showed that platelet components contribute to danger sensing and tissue repair and suggested that the interaction of platelets with leukocytes facilitates the activation of the adaptive immune response.

During adaptive immune responses, platelets promote monocyte and macrophage responses with DC and NK cell maturation, resulting in specific T- and B-cell responses. Thus, platelet granular constituents directly affect adaptive immunity by expressing CD40L [121], a molecule critical to the modulation of adaptive immune responses. Platelets via CD40L not only play a role in antigen presentation but also influence T-cell responses. Liu et al. [71] found that platelets regulate CD4 T cell responses in a complex manner. This differential regulation of CD4 T-cell subsets implies that platelets promote CD4 T cells in response to inflammatory stimuli for robust pro and anti-inflammatory responses [122].

Platelets also modulate B cell-mediated adaptive responses to microbial pathogens. It is well established that CD40L on activated CD4 T cells triggers B-cell CD40, providing the second signal necessary for T cell-dependent B-lymphocyte activation, subsequent isotype switching, and B-cell differentiation and proliferation [123]. Collectively, the results clearly indicate the various roles of platelets in adaptive immunity, suggesting that platelets augment T cell-dependent B-cell responses by linking T-cell and B-cell interactions via CD40-CD40L. Moreover, platelets have an abundance of cell surface receptors that can prompt platelet activation, with the release of numerous inflammatory and bioactive molecules stored within different platelet granules, thus influencing both innate and adaptive immune responses [124].

### 6.3. Expanded Role of Platelet-Derived Serotonin in PRP

Serotonin (5-hydroxytryptamine, 5-HT) has well-defined critical roles in the central nervous system (CNS), including pain tolerance. It is estimated that the majority of the body’s 5-HT is made in the gastrointestinal tract and then circulated by the bloodstream where it is taken up by platelets via the serotonin reuptake transporter and stored in the dense granules at high concentrations (65 mmol/L) [125]. 5-HT is a well-known neurotransmitter and hormone, contributing to the regulation of various neuropsychological processes in the CNS (central 5-HT). However, most 5-HT is found outside the CNS (peripheral 5-HT), where it is involved in regulating systemic and cellular biological functions in multiple organ systems, including the cardiovascular, pulmonary, gastrointestinal, genitourinary, and platelet functional systems [126]. 5-HT has concentration-dependent metabolic effects on diverse cell types, including adipocytes, epithelial cells, and leukocytes [127]. Peripheral 5-HT is also a powerful immune modulator that can stimulate or inhibit inflammation and affect various immune cells through their specific 5-HT receptors (5HTR) [69].

#### HT Paracrine and Autocrine Mechanisms

5-HT activities are mediated through its interaction with 5HTRs, a superfamily with seven members (5-HT_1–7_) and at least fourteen diverse receptor subtypes, including the most recently identified member, 5-HT_7_, which is expressed in the periphery and functions in pain processing [128]. During platelet degranulation, activated platelets secrete a significant amount of platelet-derived 5-HT that promotes vasoconstriction and stimulates activation of neighboring platelets and lymphocytes through the 5HTRs expressed on endothelial, smooth muscle, and immune cells. Pakala et al. [129] studied 5-HT mitogenic effects on vascular endothelial cells and identified the potential for the growth-promoting effects on damaged blood vessels by stimulating angiogenesis. How these processes are regulated is still not completely clear but presumably involves differential bi-directional signaling pathways within a tissue microenvironment to regulate the function of vascular endothelial and smooth muscle cells, fibroblasts, and immune cells through specific 5-HT receptors on these cells. The autocrine effects of platelet 5-HT after platelet activation have been described [REF]. The released 5-HT augments platelet activation and the recruitment of circulating platelets, leading to the activation of a signaling cascade and upstream effectors that support platelet reactivity [130].

### 6.4. Immunomodulatory 5-HT Effects

Accumulating evidence points to the role of serotonergic components as immunomodulators working through the different 5HTRs. Depending on the 5HTR expressed in various leukocytes involved in the inflammatory response, platelet-derived 5-HT acts as an immune regulator in both the innate and adaptive immune systems [131]. 5-HT can stimulate Treg proliferation and regulate the function of B cells, natural killer cells, and neutrophils through the recruitment of DCs and monocytes to the inflammatory sites [132,133]. Several recent studies suggest that under specific conditions, platelet-derived 5-HT can modulate immune cell functions. Therefore, the use of C-PRP, with platelet concentrations greater than 1 × 10^6^/µL, could significantly contribute to the delivery of large platelet-derived 5-HT concentrations to tissue sites. In a microenvironment characterized by inflammatory components, the PRP could interact with several immune cells that play key roles in these pathologies, potentially affecting clinical outcomes (Figure 5).

## 7. PRP Analgesic Effects

Activated platelets release many pro- and anti-inflammatory mediators that are proficient in inducing pain but can also reduce inflammation and pain. Once applied, the typical platelet dynamics of PRP alters the microenvironment prior to tissue repair and regeneration via multiple complex pathways related to anabolic and catabolic processes, cell proliferation, differentiation, and stem cell regulation. These PRP characteristics have led to the implementation of PRP applications in various clinical pathological conditions that are usually associated with chronic pain (e.g., sports injuries, orthopedic pathologies, spinal disorders, and complex chronic wounds), even though the exact mechanisms are not yet fully understood.

In 2008, Everts et al. [134] were the first to report a randomized controlled trial on the analgesic effects of a PRP formulation prepared from autologous buffy coat and activated with autologous thrombin following shoulder surgery. They noticed a significant reduction in visual analog scale scores, the use of opioid-based pain medication, and a more successful post-surgical rehabilitation. Of note, they reflected on the analgesic effects of activated platelets and postulated on the mechanism of platelet-released 5-HT. Briefly, platelets are dormant in freshly prepared PRP. After direct or indirect (tissue factor) platelet activation, platelets change shape and develop pseudopods to promote platelet aggregation. Subsequently, they release their intracellular α- and dense granules [9]. Tissues treated with activated PRP will be invaded by PGFs, cytokines, and other platelet lysosomes. More specifically, when the dense granules release their contents, an abundance of pain-modulating 5-HT will be discharged [135]. In C-PRP, the platelet concentration is 5 to 7-fold higher than in peripheral blood. Therefore, the release of 5-HT from the platelet is astronomical. Interestingly, Sprott et al. [136] reported observing substantial pain reduction following acupuncture and a significant decrease in platelet-derived 5-HT concentrations, with a subsequent increase in 5-HT plasma levels.

In the periphery, endogenous 5-HT is released from platelets, mast cells, and endothelial cells in response to tissue injury or surgical trauma, [137]. Interestingly, multiple neuronal 5-HT receptors have been detected in the periphery, confirming that 5-HT can interfere with nociceptive transmission at peripheral sites [138,139]. These studies indicate that 5-HT can affect nociceptive transmission at peripheral tissue sites through the 5-HT1, 5-HT2, 5-HT3, 5-HT4, and 5-HT7 receptors.

The 5-HT system represents a powerful system that can decrease and increase the magnitude of pain following noxious stimulation. Both central and peripheral regulation of the nociceptive signal and alterations in the 5-HT system have been reported in chronic pain patients. In recent years, considerable research has focused on the role of 5-HT and its respective receptors in the processing and modulation of noxious information [140], which led to drugs such as the selective serotonin reuptake inhibitor (SSRI). This drug inhibits the reuptake of serotonin into presynaptic neurons after serotonin has been released. It affects the duration and intensity of the serotonin communication and is an alternative treatment for chronic pain [141]. Further clinical studies are warranted to clearly understand the molecular mechanisms of the PRP-derived 5-HT pain modulatory effects in chronic and degenerative pathologies.

Additional data addressing the potential PRP analgesic effect became available following analgesic animal model trials [142]. Comparative statistical conclusions in these models were challenging because too many variables were included in these studies. Nevertheless, several clinical studies have addressed the nociceptive and analgesic effects of PRP. Several studies have indicated little to no pain relief in patients treated for tendinosis pathologies or rotator cuff tears [143,144]. In contrast, several other studies indicated that PRP reduced or even eliminated pain in patients suffering from tendinosis, OA, plantar fasciitis, and other foot and ankle disorders [145,146]. The final platelet concentration and the biocellular composition have been identified as key PRP characteristics that contributed to the consistent analgesic effects observed after PRP applications. Other variables included PRP delivery methods, application techniques, platelet activation protocols, the bioactivity levels of the released PGFs and cytokines, the types of tissues to which PRP was applied, and the type of injury.

Notably, Kuffler addressed the potential of PRP in pain relief in patients suffering from mild to severe chronic neuropathic pain, secondary to a damaged non-regenerated nerve. The objective of this study was to investigate whether neuropathic pain would decrease or resolve as a result of PRP’s promotion of axonal regeneration and target reinnervation [147]. Strikingly, in treated patients, the neuropathic pain remained eliminated, or reduced, for a minimum of six years after the procedure. Furthermore, pain started to decrease within three weeks after the surgical PRP application in all patients.

Recently, similar analgesic PRP effects were observed in the field of post-surgical wound and skincare. Interestingly, the authors reported the physiological aspects of wound pain related to vascular injury and skin tissue hypoxia. They also discussed the importance of neoangiogenesis in optimizing oxygenation and nutrient delivery. Their study demonstrated pain reduction in PRP-treated patients compared to controls and significantly higher angiogenesis in the PRP-treated patients [148]. Finally, Johal and co-workers performed a systematic review and meta-analysis and concluded that PRP leads to a reduction in pain following PRP administration in orthopedic indications, particularly in patients treated for lateral epicondylitis and knee OA [12]. Unfortunately, this study did not specify the effects of leukocytes, platelet concentration, or the use of exogenous platelet-activating agents, as these variables affect the overall PRP effectiveness. The optimal PRP platelet concentration that provokes maximal pain relief is yet unknown. In a rat tendinopathy model, complete pain relief was accomplished with a platelet concentration of 1.0 × 10^6^/µL, whereas PRP with half this platelet concentration induced significantly less pain relief [142]. Thus, we encourage more clinical studies to investigate the analgesic effects of different PRP formulations.

## 8. PRP and Angiogenesis Effects

C-PRP preparations in precision regenerative medicine therapies allow for the delivery of biomolecules released by the high concentrations of platelets activated at the target tissue sites. As a result, various cascades are initiated that contribute to on-site immunomodulation, inflammatory processes, and angiogenesis to promote healing and tissue repair [149].

Angiogenesis is a vibrant, multistep process involving the sprouting and organization of microvessels from pre-existing blood vessels. Angiogenesis progresses due to multiple biological mechanisms, including endothelial cell migration, proliferation, differentiation, and division. These cellular processes are prerequisites to the formation of new blood vessels. They are essential for the outgrowth of pre-existing blood vessels to restore blood flow and support the high metabolic activity of tissue repair and tissue regeneration. These new vessels allow the delivery of oxygen and nutrients and the removal of byproducts from the treated tissues [70].

Angiogenic activities are modulated by the stimulatory pro-angiogenic factor VEGF and anti-angiogenic factors (e.g., angiostatin and thrombospondin-1 [TSP-1]). Within a diseased and degenerative microenvironment (including a low oxygen tension, low pH and high lactate levels), local angiogenic factors restore angiogenic activities. Several platelet soluble mediators, such as basic-FGF, TGF-β, and VEGF, stimulate endothelial cells to produce new blood vessels [70]. Landsdown and Fortier [150] reported on the various outcome effects related to the PRP constituents, including intra-platelet sources of numerous angiogenic modulators. Furthermore, they concluded that an increase in angiogenesis contributes to the healing of MSK disorders in areas of poor vascularization, such as meniscal tears, tendon injuries, and other areas with poor vascularity.

### Pro- and Anti-Angiogenic Platelet Properties

During the last decades, published studies demonstrated the pivotal role of platelets in primary hemostasis, clot formation, growth factor and cytokine release, and the regulation of angiogenesis as part of the tissue repair process. Paradoxically, PRP contains an armory of both pro-angiogenic growth factors and anti-angiogenic proteins and cytokines (e.g., PF4, plasminogen activation inhibitor-1, and TSP-1) in the α-granules, targeting the release of specific factors that play a role in blood vessel formation. Therefore, the role of PRP in controlling angiogenic modulation may be defined by the activation of specific cell surface receptors, with TGF-β eliciting both pro and anti-angiogenic responses, as shown in Table 4 [151]. The capability of platelets in exercising angiogenic pathways has been demonstrated in pathological angiogenesis [152] and tumor angiogenesis [153].

Most importantly, the well-accepted opinion is that the overall platelet effect on angiogenesis is pro-angiogenic and stimulatory [154]. The controlled induction of angiogenesis is anticipated with PRP therapies, which will contribute to the therapeutic efficacy for a number of conditions, such as wound healing and tissue restoration. The administration of PRP, more specifically the delivery of high concentrations of PGFs and other platelet cytokines can induce angiogenesis, vasculogenesis, and arteriogenesis because stromal cell-derived factor-1a binds to the CXCR4 receptors on endothelial progenitor cells. Bir et al. [155] showed that PRP augments ischemic neovascularization, presumably due to the stimulation of angiogenesis, vasculogenesis, and arteriogenesis. In their in vitro model, endothelial cell proliferation and capillary tube formation were induced by the large amounts of different PDGs, where VEGF was the principle angiogenic stimulatory factor. Another important and essential factor in restoring angiogenic pathways is synergy between multiple PGFs. Richardson et al. [156] demonstrated that the synergistic activities of the angiogenic factors platelet derived growth factor-bb (PDGF-BB) and VEGF results in the rapid formation of a mature vascular network compared to the individual growth factor activities. The combined activity of these factors was recently confirmed in a study addressing the augmentation of collateral circulation in the chronically hypo-perfused brain in mice [157].

Most importantly, an in vitro study measuring the proliferative effects of human umbilical vein endothelial cells and various platelet concentrations with regard to the choice of PRP preparation devices and platelet dosing strategies revealed that the optimal platelet dose is 1.5 × 10^6^ platelets/µL to promote angiogenesis. Excessive platelet concentrations might inhibit the angiogenic process and, thus, be less effective [65].

## 9. Cell Senescence, Aging, and PRP

Cellular senescence can be induced by a variety of stimuli. It is a process in which cells cease dividing and undergo distinctive phenotypic alterations guarding against unrestricted growth of damaged cells, playing important roles as a safeguard against cancer [158]. During physiological aging, cellular senescence can also be prompted upon replicative aging of cells, with a decreased regenerative capacity of MSCs [159].

### 9.1. Effects of Aging and Cell Senescence

In vivo, a number of cell types can become senescent and accumulate in various tissues when aging [160,161] where senescent cell can be abundantly present. It appears that senescent cell accumulation increases with age, in a compromised immune system, tissue damage, or stress related factors. Cell senescence mechanisms have been identified as causative factors in age-related diseases like e.g., osteoarthritis, osteoporosis, and vertebral disc degeneration [158,162,163,164]. Multiple stimuli can aggravate cellular senescence and in response a senescence-associated secretory phenotype (SASP) secretes high concentrations of proteins cells and cytokines. This particular phenotype is associated with senescent cells wherein they secrete high levels of inflammatory cytokines (e.g., IL-1, IL-6, IL-8), growth factors (e.g., TGF-β, HGF, VEGF, PDGF), MMPs, and cathepsins [165]. SAPS has been shown to increase with age, as homeostatic processes are disrupted leading to cell senescence and reduced regenerative capacity when compared to younger adults. Specifically, in joint pathologies and skeletal muscle disorders. In this regard, immuno-senescence has been recognized as a significant alteration in the secretory profile of immune cells demonstrating increased concentrations of TNF-a, IL-6, and/or Il-1b, inducing low grade chronic inflammation [166]. Noteworthy, stem cell dysfunction is also associated with non-cell autonomous mechanisms, like senescent cells, in particular through SASP with the production of associated pro-inflammatory and anti-regenerative factors [167].

Conversely, SASP can also stimulate cell plasticity and reprogramming in adjacent cells. Furthermore, SASP can organize communication with various immune mediators, enabling immune cell activation to facilitate senescent cell clearance [168]. Understanding the roles and functions of senescent cells can contribute to healing and tissue remodeling in MSK muscles and chronic wounds [169].

Notably, Ritscka et al. executed an extensive study and uncovered a primary and beneficial role for SASP in promoting cell plasticity and tissue pro-regenerative responses, introducing the concept that transient therapeutic delivery of senescent cells. They cautiously mention the idea that senescence is primarily a beneficial and regenerative process [160].

### 9.2. Cell Senescence and the Potential of PRP

Aging has been recognized to affect stem cell performance as their numbers decline [170]. Similarly, in humans, stem cell characteristics like stemness, proliferation, and differentiation also decrease with age [171]. Wang and Nirmala reported that aging diminishes the properties of tenocyte stem cells and the number of growth factor receptors were decreased. An animal study revealed that PDGF concentrations were higher in younger horses [169]. They concluded that the increased number of GF receptors and higher amounts of GFs in young individuals may trigger a better cellular response to PRP treatment in young than older individuals. These findings shed some insights on why PRP treatment may be less effective or even ineffective in older patients who may have fewer numbers and “poor quality” of stem cells [172]. Reversal of the senescence processes in aging cartilage following PRP injections has been demonstrated, with an increase in chondrocyte quiescence [173]. Jia et al. used in an in-vitro photoaging study murine dermal fibroblasts, treated with and without PRP to elucidate the counteracting mechanisms of PGFs in this model. The PRP group demonstrated direct effects on the extracellular matrices, increased type I collagen and decreased metalloproteinase synthesis, suggesting that PRP can counteract cell senescence, also in degenerative MSK disorders [174].

In another study, PRP was used in senescent bone marrow stem cells harvested from aged mice. It was determined that PRP was capable of recovering several stem cell functions from senescence, such as cell proliferation and colony formation, while reestablishing cell senescence related markers [171].

Recently, Oberlohr and associates studied extensively the role of cell senescence in attenuated muscle regeneration, evaluating PRP and platelet poor plasma (PPP) as biological treatment options for skeletal muscle repair. They envisioned that PRP or PPP treatment for skeletal muscle repair will be based on the customization of biological factors by targeting SASP specific cell markers and other factors that attribute to the development of fibrosis [175].

It stands to reason that targeting cell senescence may improve the regenerative properties of biological treatment efficacy through the reduction of local SASP factors prior to PRP applications. Another option that has been suggested to improve PRP and PPP treatment outcomes for skeletal muscle regeneration is the use of senolytic agents to selectively remove senescent cells [176]. Without a doubt, recent research findings of PRP effects on cell senescence and aging are fascinating, but still in its infancy. Therefore, it is unreasonable to make any recommendations at this moment.

## 10. The Role of Platelets in Bone Marrow Aspirate Concentrate

PRP and bone marrow aspirate concentrates (BMACs) are being used in a range of clinical treatments in office settings and surgical procedures for their regenerative benefits in MSK and spinal disorders, chronic pain management, and soft tissue indications. PRP components not only regulate cell migration and cell proliferation but also contribute to angiogenesis and the remodeling of the ECM to create a favorable microenvironment that enhances tissue repair and regeneration.

### 10.1. BMAC Repair Processes

BMACs are heterogeneous cell compositions that include BMMSCs, making them endogenous cell sources for regenerative medicine repair treatments. They act by reducing cell apoptosis, fibrosis, and inflammation; and activating cascades that lead to cell proliferation. In addition, BMMSCs have the potential to differentiate into multiple cell lineages, including osteoblasts, adipocytes, myoblasts, epithelial, and neuronal cells. They also contribute to angiogenesis via paracrine and autocrine pathways [177]. Equally important, BMMSCs are contributors to immunomodulatory actions independent of immune-specific cells, which participate in the inflammatory phase of wound repair. Moreover, BMMSCs support the recruitment of cells to neoangiogenic treatment sites to accelerate local revascularization [10]. Kim et al. demonstrated that in the absence of an adequate scaffold, the survival rate of BMMSCs and their reparative and differentiation capacity to enhance healing are jeopardized [178]. Although tissue harvesting, specimen preparation, and mechanism of action are different for PRP and BMACs, studies have shown that they can complement each other [179,180]. Indeed, there may be an added advantage to combining PRP and BMACs into one biological product.

### 10.2. Combining PRP and BMACs

The rationale to combine PRP and BMAC is based on several premises, according to some scarcely available studies. First, the ability of PRP to provide a suitable microenvironment in which BMSCs can augment cell proliferation and differentiation and increase neoangiogenesis [181]. Secondly, PRP has been used together with BMAC to act as a scaffold for these cells. Conversely, PRP combined with BMAC can be a powerful biological tool to attract BMMSC populations. PRP-BMAC complexes have been used to treat tendinopathies, wounds, spinal cord injuries, degenerated discs, and osteochondral defects with great regenerative potential [179,182]. Unfortunately, there are not many reports that mention platelet concentrations in extracted bone marrow and after BMAC processing, albeit that the heterogenous bone marrow cellular composition includes platelets, which can be extracted with proper aspiration methods [10]. Further research is warranted to understand the need for additional platelet concentrates to be combined with BMAC. Currently, no data exist on the optimal platelet to MSC (or other bone marrow cells) cell ratio, leading to a positive effect on the MSC trophic mechanisms in tissue repair. Ideally, bone marrow harvesting devices and techniques can be optimized in order to extract a sufficient number of bone marrow platelets.

### 10.3. PRP Growth Factors and BMAC Trophic Effects

PRP platelet growth factors are crucial proteins that are involved in the BMAC reparative processes. The diversity of PGFs and other cytokines involved in BMAC trophic processes can initiate tissue repair by decreasing cell apoptosis, anabolic and anti-inflammatory effects, and activating cell proliferation, differentiation, and angiogenesis via paracrine and autocrine pathways, as presented in Figure 6 [183,184].

Explicitly in OA treatments, PDGF plays a specific role in cartilage regeneration and maintaining homeostasis via MSC proliferation and the inhibition of IL-1-induced chondrocyte apoptosis and inflammation [185]. In addition, three TGF-β isoforms are active in stimulating chondrogenesis, inhibiting inflammation, and they express their ability to promote MSC associated tissue healing via inter-molecular actions [183]. MSC trophic effects are associated with PGF activity and the secretion of reparative cytokines [177]. Ideally, all of these cellular factors should be present in the BMAC treatment vials and delivered to tissue injury sites to promote optimal MSC-associated therapeutic tissue healing [184].

In a joint OA study, Muiños-López et al. [186] showed that MSCs derived from synovial tissues have altered function, resulting in the loss of their restorative capacity. Interestingly, PRP injections directly into the osteoarthritic subchondral bone caused a decrease in MSCs in the synovial fluid, suggesting clinical improvement. The therapeutic effects are mediated by a decrease in the pro-inflammatory processes present in the synovial fluid of OA patients.

There is minimal information available on the presence or concentrations of PGFs in BMACs, or the ideal ratio needed to support BMMSC trophic actions. Some clinicians combine high PRP concentrations with BMACs to have potentially a more biologically active graft, projected to optimize regenerative medicine treatment outcomes [180,187]. However, there are minimal safety and efficacy data available that indicates that combining high PRP concentrations with BMAC is a more effective treatment option. Therefore, we believe that manipulating BMMSCs by priming them with high platelet concentrations may not be indicated at this stage.

## 11. Platelet Interactions with Anti-Platelet Medications and NSAIDs

PRP contains a broad secretome profile consisting of many biological mediators [188]. The therapeutic effects of PRP are attributed to these mediators. Although the therapeutic mediators within platelets are well-known, the optimal formulation and kinetics of these anabolic and catabolic agents are not fully understood. One of the main limitations in achieving therapeutic formulations is overcoming the variability of these biological mediators to target well-regulated downstream effects that are consistently reproducible and clinically beneficial. To this end, medications (e.g., non-steroidal anti-inflammatory drugs (NSAIDs) can affect platelet secretome release. In a recent open-label fixed sequence study, a daily intake of 81 mg aspirin (ASA) reduced the expression of key mediators, such as TGF-β1, PDGF, and VEGF [189]. These effects were attributed to irreversible inhibition of cyclooxygenase-1 (COX-1) and modifiable inhibition of cyclooxygenase-2 (COX-2), two enzymes needed for downstream platelet degranulation. A recent systematic review found that antiplatelet medications may decrease the growth factor release profile in a COX-1 and COX-2 dependent manner, and eight of the 15 studies found a decrease in growth factors [190].

Pharmaceutical agents (e.g., NSAIDs) are often used to ease pain and reduce inflammation from MSK disorders. The mechanism of action of NSAIDs is to inhibit platelet activation by irreversibly binding to the COX enzymes and modulating the arachidonic acid pathway [191]. Consequently, platelet function is altered across the platelet’s life span, preventing PGF signaling [192]. NSAIDs inhibit cytokine production (e.g., PDGF, FGF, VEGF, and interleukins IL-1b, IL-6, and IL-8), while enhancing TNF-α [193,194]. However, data regarding the molecular influence of NSAIDs on PRP is scarcely available. A consensus on the best time for PRP preparation and administration in patients who use NSAIDs is lacking. Mannava and associates quantified the anabolic and catabolic biological factors in leukocyte-rich PRP from healthy volunteers taking naproxen [195]. They found that the levels of PDGF-AA and PDGF-AB, potent mitogens that promote angiogenesis, were significantly reduced after one week of naproxen use. After a week of washout, the growth factor levels returned to near baseline levels. The pro-inflammatory catabolic factor IL-6 also showed diminished levels in LR-PRP after one week of naproxen use, which returned to baseline levels after a one-week washout period. There are currently no clinical studies demonstrating negative patient outcomes following naproxen use post-PRP treatment; however, it is advisable to consider a one-week washout period to allow PDGF-AA, PDGF-BB, and IL-6 values return to baseline levels to improve their biological activity. More studies are needed to comprehensively understand antiplatelet and NSAID effects on the PRP secretome and their downstream targets.

## 12. Combining Platelet-Rich Plasma Applications with Rehabilitation

There is no consensus on optimal rehabilitation protocols after PRP treatment for MSK disorders, even though basic science studies suggest clear roles for physical therapy and mechanical loading in the restoration of tendon structure post-PRP injections [196].

PRP treatment involves injecting concentrated platelets in the local tissue milieu to modulate pain and foster tissue repair [197,198]. The strongest clinical evidence exists in knee OA. However, the use of PRP for symptomatic tendinopathy is controversial, with mixed results reported. Animal studies generally show histologic improvement following PRP infiltration for tendinopathy [199]. These studies showed that mechanical loading is regenerative to tendons, and the load is synergistic with PRP injections for tendon healing. Variations in PRP preparation, bioformulation, preparation, injection protocols, and tendon injury subtypes may account for the variability in clinical outcomes [199]. Furthermore, despite scientific evidence supporting the benefit of rehabilitation programs, very few published clinical investigations have attempted to manage and incorporate consistent post-PRP rehabilitation programs [200].

Recently, Onishi et al. [196] reviewed both the role of mechanical loading and PRP biological effects in Achilles tendinopathy. They evaluated Phase I and II clinical studies for PRP-treated Achilles tendinopathy, focusing on post-PRP injection rehabilitation programs. Supervised rehabilitation programs appeared to increase exercise compliance and improve outcomes and the ability to monitor exercise dosing [201,202]. Several well-designed Achilles tendon PRP trials integrated post-PRP treatment with a mechanical loading rehabilitation program as an integrated part of the regenerative strategy.

## 13. Future Prospect and Conclusions

Technological advances in PRP devices and preparation methodologies show promising patient outcome results, although clarity in different PRP bioformulations and the related biological properties of the final product is still not conclusive. Moreover, the full potential of PRP indications and applications has yet to be determined. Until recently, PRP has been commercially marketed as an autologous blood-derived product potentially offering physicians the ability to use autologous platelet growth factor technologies in specific indicated pathologies and disorders. Originally, the frequently cited sole criterion for successful PRP applications was a prepared specimen with a platelet concentration above the whole blood value. Today, fortunately, practitioners have a more comprehensive understanding on the modus operandi of PRP.

In this review, we acknowledged that there is still a lack in standardization and classification regarding preparation techniques; consequently, no consensus on PRP bio-formulations is existing, albeit that more literature is in agreement regarding effective platelet dosing concentrations needed to facilitate (neo)angiogenesis. Here, we touched briefly on the activity of PGFs, but reflected more extensively on specific platelet mechanisms and effector actions regarding leukocytes, MSCs, with consequential cell-cell interactions. In particular, the presence of leukocytes in PRP preparations provide greater insights into either deleterious or beneficial effects. Explicit roles for platelets and their interactions with both the innate and adaptive immune system have been deliberated. Furthermore, we addressed the functionality of PRP in inflammation and pain killing and the combination of PRP with aspirin and NSAIDs, indicating a reduction in growth factor release, potentially affecting patient outcomes.

Sufficiently powered and well-documented clinical studies are needed to determine the full potential and thus therapeutic effects of PRP, in a variety of indications.

## Figures and Tables

**Figure 1 ijms-21-07794-f001:**
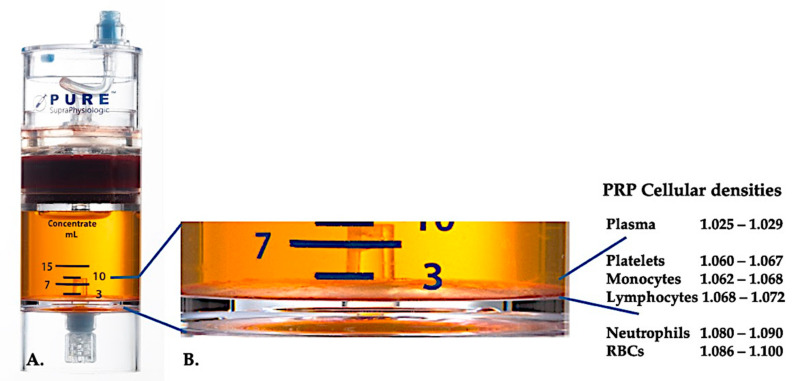
Cellular density separation of whole blood following a two-spin centrifugation procedure using the PurePRP-SP^®^ device (EmCyte Corporation, Fort Myers, FL, USA). After the first centrifugation procedure, the whole blood components are separated in two basic layers, the platelet (poor) plasma suspension and the RBC layer. In A, the second centrifugation step has been completed. The factual needed PRP volume can be extracted for patient application. The magnification in B shows at the bottom of the device the organized multicomponent buffy coat layer (indicated by the blue lines), containing high concentrations of platelets, monocytes, lymphocytes, based on density gradients. In this example, a minimal percentage of neutrophils (<0.3%) and RBCs (<0.1%) will be extracted, following a neutrophil poor C-PRP preparation protocol.

**Figure 2 ijms-21-07794-f002:**
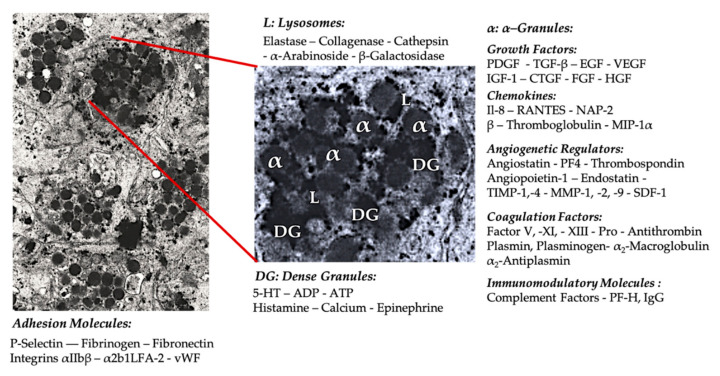
Electron microscopic picture of a cluster of platelets from a PRP vial and a extrapolation of a single platelet (original magnification × 10,000) (from volunteer PE), representing the most familiar cellular constituents of α-granules (α), dense granules (DG), and lysosomes (L), including some platelet surface adhesion molecules. Adapted and modified from Everts et al. [61].

**Figure 3 ijms-21-07794-f003:**
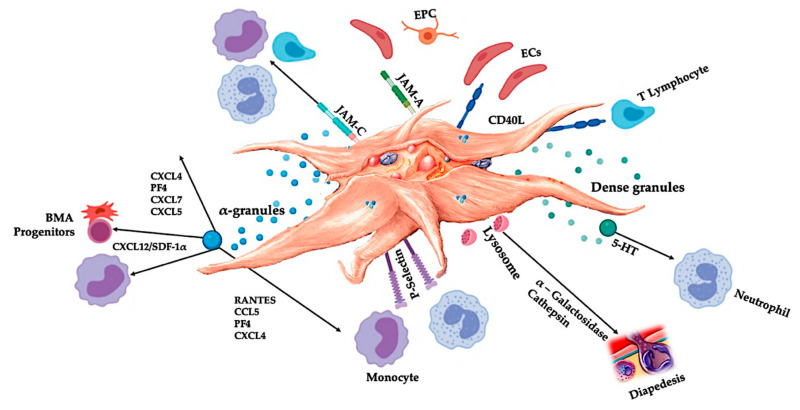
Activated platelets, releasing PGF, and adhesion molecules mediate a variety of cellular interactions: chemotaxis, cell adhesion, migration, cell differentiation, and stipulate to immunomodulatory activities [67,68]. These platelet cell-cell interactions contribute to angiogenesis [46,69,70] and inflammatory [71,72] activities, ultimately to stimulate tissue repair processes. Abbreviations: BMA: bone marrow aspirate, EPC: endothelial progenitor cell, EC: endothelial cells, 5-HT: serotonin, RANTES: Regulated upon Activation Normal T Cell Expressed and Presumably Secreted, JAM: junctional adhesion molecules type, CD40L: cluster of differentiation 40 ligand, SDF-1α: stromal cell-derived factor 1 alpha, CXCL: chemokine (C-X-C motif) ligand, PF4: platelet factor 4. Adapted and modified from Everts et al. [9].

**Figure 4 ijms-21-07794-f004:**
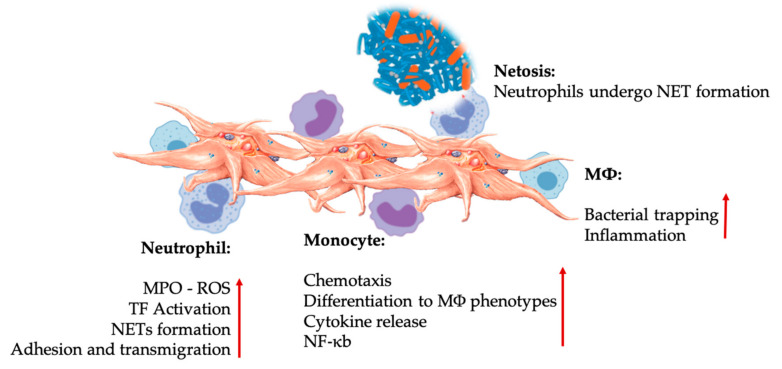
Platelet and leukocyte interactions in innate immunity cell interactions. Platelets interact with neutrophils, monocytes, and ultimately as well with MΦs, modulating and increasing their effector functions. These platelet-leukocyte interactions result in inflammatory contributions through different mechanisms, including NETosis [67]. Abbreviations: MPO: myeloperoxidase, ROS: reactive oxygen species, TF: tissue factor, NET: neutrophil extracellular traps, NF-κB: nuclear factor kappa B, MΦ: macrophage.

**Figure 5 ijms-21-07794-f005:**
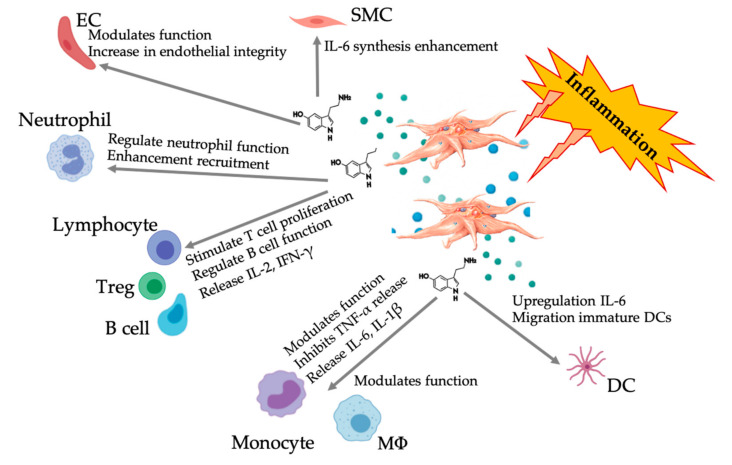
Illustration of the multifaceted 5-HT responses following inflammatory PRP-platelet activation. After platelet activation, platelets release their granules, including 5-HT from dense granules, inciting a wide range of differential effects on various immune, endothelial, and smooth muscle cells. Abbreviations: SMC: smooth muscle cell, EC: endothelial cell, Treg: regular T lymphocyte, MΦ: macrophage, DC: dendritic cell, IL: interleukin, IFN-γ: interferon gamma. Modified and adapted from Everts et al. and Herr et al. [9,69].

**Figure 6 ijms-21-07794-f006:**
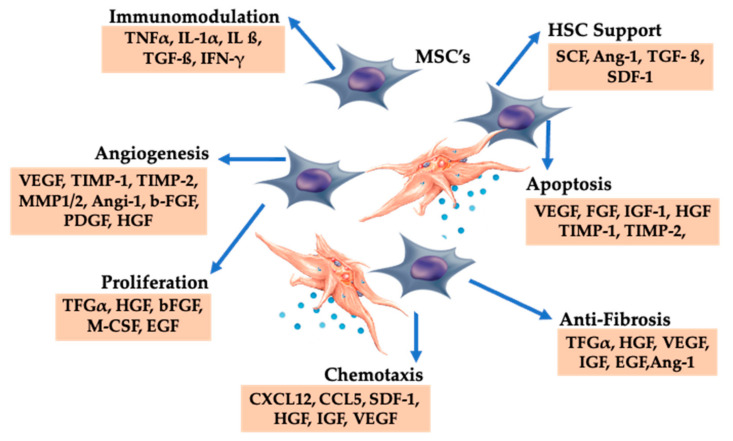
Platelet-derived growth factors and dense granular constituents are expressively involved in BMAC trophic processes, supporting MSC induced tissue repair and regeneration. Abbreviations: MSC: mesenchymal stem cell, HSC: hematopoietic stem cell.

**Table 1 ijms-21-07794-t001:** Partial list of PRP based growth factors and platelet cytokines with their cell sources.

PGF and Cytokines	Cell Sources	Function and Effects
PDGF (AA-BB-AB)	Platelets, endothelial cells, macrophages, smooth muscle cells	Mitogenic for mesenchymal cells and osteoblasts; stimulates chemotaxis and mitogenesis in fibroblast/ glial/smooth muscle cells; regulates collagenase secretion and collagen synthesis; stimulates macrophage and neutrophil chemotaxis
TGF (α–β)	Macrophages, T lymphocytes, keratinocytes	Stimulates undifferentiated mesenchymal cell proliferation; regulates endothelial, fibroblastic, and osteoblastic mitogenesis; regulates collagen synthesis and collagenase secretion; regulates mitogenic effects of other growth factors; stimulates endothelial chemotaxis and angiogenesis; inhibits macrophage and lymphocyte proliferation
VEGF	Platelets, macrophages, keratinocytes, endothelial cells	Increases angiogenesis and vessel permeability; stimulates mitogenesis for endothelial cells
EGF	Platelets, macrophages, monocytes	Proliferation of keratinocytes, fibroblasts, stimulates mitogenesis for endothelial cells
(a-b)-FGF	Platelets, macrophages, mesenchymal cells, chondrocytes, osteoblasts	Promotes growth and differentiation of chondrocytes and osteoblasts; mitogenic for mesenchymal cells, chondrocytes, and osteoblasts
CTGF	Platelets, fibroblasts	Promotes angiogenesis, cartilage regeneration, fibrosis, and platelet adhesion
IGF-1	Platelets, plasma, epithelial cells, endothelial cells, fibroblasts, osteoblasts, bone matrix	Chemotactic for fibroblasts and stimulates protein synthesis. Enhances bone formation by proliferation and differentiation of osteoblasts
HGF	Platelets, mesenchymal cells	Regulates cell growth and motility in epithelial/endothelial cells, supporting epithelial repair and neovascularization during wound healing
KGF	Fibroblasts, mesenchymal cells	Regulates epithelial migration and proliferation
Ang-1	Platelets, neutrophils	Induces angiogenesis stimulating migration and proliferation of endothelial cells. Supports and stabilizes blood vessel development via the recruitment of pericyte
PF4	Platelets	Calls leucocytes and regulates their activation. Microbiocidal activities
SDF-1α	Platelets, endothelial cells, fibroblasts	Calls CD34+ cells, induces their homing, proliferation and differentiation into endothelial progenitor cells stimulating angiogenesis. Calls mesenchymal stem cells and leucocytes
TNF	Macrophages, mast cells, T lymphocytes	Regulates monocyte migration, fibroblast proliferation, macrophage activation, angiogenesis

Modified from Everts et al. [6] and Giusti et al. [13]. Abbreviations: PDGF: platelet-derived growth factors; TGF: transforming growth factor; VEGF: vascular endothelial growth factor; EGF: epidermal growth factor; FGF: fibroblast growth factor; CTCG: connective tissue growth factor; IGF: insulin-like growth factor; HGF: hepatocyte growth factor; KGF: keratinocyte growth factor; Ang-1: angiopoietin-1; PF4: platelet factor 4; SDF: stromal cell derived factor; TNF: tumor necrosis factor.

**Table 2 ijms-21-07794-t002:** PRP product related terminologies and their abbreviations.

A-PRF	Advanced Platelet-Rich Fibrin
ACP	Autologous Conditioned Plasma
AGF	Autologous Growth Factors
APG	Autologous Platelet Gel
C-PRP	Clinical Platelet-Rich Plasma
i-PRF	Injectable Platelet-Rich Fibrin
LP-PRP	Leukocyte-Poor Platelet-Rich Plasma
LR-PRP	Leukocyte-Rich Platelet-Rich Plasma
PFC	Platelet-derived Factor Concentrate
P-PRP	Pure Platelet Rich Plasma
PFS	Platelet Fibrin Sealant
PLG	Platelet-Leukocyte Gel
PRF	Platelet-Rich Fibrin
PRFM	Platelet-Rich Fibrin Matrix
PRGF	Preparation Rich in Growth Factors

**Table 3 ijms-21-07794-t003:** Parameters to be considered in developing a PRP classification system.

Parameters	Differentials	Options
Biological Product Allocation	Autologous Allogeneic	Buffy Coat Partial Buffy Coat Fresh Frozen/Thawed Platelet Lysate Umbilical cord blood
Preparation Technology	Gravitational Centrifugation Blood Salvage Blood Separators Plasmapheresis	Preparation time Spin-Cycles G-Forces
Anticoagulation	ACD-A EDTA SC Heparin	
Platelet dosing	Concentration ranges	0–500 × 10^6^/mL 500–1000 × 10^6^/mL 1000–1500 × 10^6^/mL >1500 × 10^6^/mL
Leukocytes Presence	Yes No	Neutrophils–Monocytes–Lymphocytes Poor–Poor
RBC	Yes No	Hematocrit (range)
Delivery Form	Liquid Coagulated	Partial Full
Fibrin Matrix	Yes No	Concentration levels Content specific
Activation	Yes No	CaCl Thrombin Collagen Electrical Freeze Sonication Light
Additives	Biodegradable Scaffolds Matrices Autologous Biologics Non-autologous	Dexamethasone—HA—cPPP—BMAC—Adipose—Bone-Exosomes—Amniotic -Wharton Jelly—A-Cell Protein Preparations— Antibiotics—Pain medication
Administration Routes	Topical IV Tissue structure Intraosseous	Soft tissue: Tendon—Ligament—Muscle—Scar Intradiscal—Epidural—Intrathecal—Intra-Articular

Abbreviations: G-Force: gravitational force, ACD-A: Anticoagulant Citrate Dextrose Solution-Solution A, EDTA: Ethylenediaminetetraacetic acid, SC: sodium citrate, CaCl: calcium chloride, HA: Hyaluronic Acid, cPPP: concentrate platelet poor plasma, BMAC: bone marrow concentrate, IV: intravenous.

**Table 4 ijms-21-07794-t004:** Platelet-derived pro and anti-angiogenetic growth factors, originating from α- and dense and adhesion molecules.

Pro-Angiogenetic	Anti-Angiogenetic
VEGF	TGF-β1
PDGF	PAI
TGF-β1	TSP
EGF	Angiostatin
Serotonin	Endostatin
SDF-1	PF4
Angiopoietin -1, -2	CXCL4L
MMP -1, -2	TIMPS
IL-8

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
