# Peer review of "Platelet-Rich Plasma: New Performance Understandings and Therapeutic Considerations in 2020"

_ijms, 2020, doi:10.3390/ijms21207794_

Round 1
Reviewer 1 Report
This manuscript will provide further info about platelet-derived products and their their direct and indirect effects on the immunomodulation. Indeed, the review comprises many scientific critical point of views about the PRP.
Here, my comments and suggestion to improve the manuscript.
- About introduction. I believe that the authors found and mentioned an important issue. The standardization and bioformulation problem can be more discussed. also expanding the issue and considering the pre-clinical application of PRP-like products, for example platelet lysate used for MSCs (you can have a look at doi: 10.1155/2016/7230987). Platelet lysate may likely share heterogenicity, including some rationale, dosing problems and healing/immunological effects of the most used PRP.
- The in vitro use of PRP and "similars" is reported also at lines 232-3 by the authors; I would say that it is not "precisely", but just controlled, because the PRP formulation is even not completely clear in products for research-use-only.
- Check please that both table entries and legend abbreviate at the same manner. For example PF-4 and PF4; please use only a unique abbreviation for that growth factor. Do the same in the whole document.
- Para 3.2 and 3.4 can be joined and summarized.
- May the authors consider to mention or include in the table 3 that also the selection of the tissue for blood collection is a parameter? Which is the authors obejctive opinion about platelet-derived products (heterogenicity vs advantage) from newborn cord blood instead of adult peripheral blood? And what is known about C-PRP and cord blood? This kind of blood include also particular stem cell types (than hematopoietic linaage.
- The authors should briefly specify if there are still commercially (for both clinical and in vitro use)available products not following the advocated PRP Classification System
- I sueggest to discuss the senescence level of the blood composition (then the age of donors) and how it influence PRP preparation and efficacy.
- I think it is too much speculative and not so much informative the text from 906th line ("The lack of supervision...) to the end (919th line). Consider to remove.
- Minor comment.Check abbreviation for ... "cyclooxygenase-1 (COX-1) and modifiable inhibition of COX-2"
- The manuscript needs a conclusion paragraph. What's the expected and future outcome and indication of the review study?
Author Response
Response to Reviewer 1 Comments
Point 1: About introduction. I believe that the authors found and mentioned an important issue. The standardization and bioformulation problem can be more discussed. also expanding the issue and considering the pre-clinical application of PRP-like products, for example platelet lysate used for MSCs (you can have a look at doi: 10.1155/2016/7230987). Platelet lysate may likely share heterogenicity, including some rationale, dosing problems and healing/immunological effects of the most used PRP.
Response 1:
Thank you for your comments and suggestion.
We added the below text in line 48 and added a reference.
Furthermore, the lack of consensus on standardization of PRP preparation protocols, with adequate reporting on bioformulations in clinical applications, contributes to inconsistencies in reported outcomes. Several attempts have been made to characterize and classify PRP or blood-derived products in regenerative medicine applications. In addition, platelet-derivates, like human platelet lysate, have been proposed in orthopedics and in-vitro stem cell research.
Point 2: The in vitro use of PRP and "similars" is reported also at lines 232-3 by the authors; I would say that it is not "precisely", but just controlled, because the PRP formulation is even not completely clear in products for research-use-only.
Response 2:
Thank you for your suggestion, we changed to controlled in line 286.
Point 3: Check please that both table entries and legend abbreviate at the same manner. For example PF-4 and PF4; please use only a unique abbreviation for that growth factor. Do the same in the whole document.
Response 3:
Thank for your observation, we checked the document as per your suggestions; line 139, 809
Point 4: Para 3.2 and 3.4 can be joined and summarized.
Response 4:
Thank you for your comment. We combined paragraph 3.2 and 3.4 and summarized the new paragraph. Changes started in line 164.
Point 5: May the authors consider to mention or include in the table 3 that also the selection of the tissue for blood collection is a parameter? Which is the authors obejctive opinion about platelet-derived products (heterogenicity vs advantage) from newborn cord blood instead of adult peripheral blood? And what is known about C-PRP and cord blood? This kind of blood include also particular stem cell types (than hematopoietic linaage.
Response5:
We gave your comment addressing or adding allogeneic cord blood to table 3 considerable consideration. We added “umbilical cord blood” as a source for allogeneic PRP, thank you for the suggestion. Line 227 of table 3. Since the fous is onautologous PRP preaprations and most of the itemized options are not further discussed, but only mentioned to consider in a new classification system we feel not to follow up with further discussions, in general.
Hope that this is sufficiently explained
Point 6: The authors should briefly specify if there are still commercially (for both clinical and in vitro use) available products not following the advocated PRP Classification System
Response 6:
Thanks for your comment. Starting in line 353, we added text addressing your suggestion.
Point 7: I sueggest to discuss the senescence level of the blood composition (then the age of donors) and how it influence PRP preparation and efficacy.
Response 7:
Thank you for your suggestion. We added a new paragraph to the manuscript: “ Cell Senescence, Aging, and PRP”, including references. The inserted paragraph is starting at line 922,
Point 8: I think it is too much speculative and not so much informative the text from 906th line ("The lack of supervision...) to the end (919th line). Consider to remove.
Response 8:
We follow your reasoning and deleted the text, starting at new line 1038.
Point 9: Minor comment.Check abbreviation for ... "cyclooxygenase-1 (COX-1) and modifiable inhibition of COX-2"
Response 9:
Thank you for catching this, changes have been made accordingly in line 996 and line 998
Point 10: The manuscript needs a conclusion paragraph. What's the expected and future outcome and indication of the review study?
Response 10:
Thank you for making this comment, apologies for this omission. We added conclusions, starting line 1053.
Reviewer 2 Report
The manuscript entitled “Platelet-Rich Plasma: New Performance Understandings and Therapeutic Considerations in 2020” discussed recent developments regarding PRP preparation and composition regarding platelet dosing, leukocyte activities concerning innate and adaptive immunomodulation, serotonin (5-HT) effects and pain killing, also some mechanism. This review manuscript is nice and need minor revision.
Issues are listed below.
- Author further can improve information’s related to the PRP in all previous literature.
- Author must show previous reported molecular biology results with validated mechanisms in this manuscript not just graphical mechanism images.
- Author must describe in Figure 2 caption, whether this is their unpublished result or taken form somewhere else. If it is taken from other published source, please mention copyright detail in the caption. It must be same for all relevant figures.
- Experimental data related to immunomodulation must be added with copyright information in this manuscript.
- At the end section ‘Future Prospective and Conclusion’ must be included in this manuscript.
Author Response
Response to Reviewer 2 Comments
Point 1: Author further can improve information’s related to the PRP in all previous literature.
Response 1:
We are sorry Sir, we do not understand your suggestion here. Please elaborate if you don’t mind, in order for us to respond in a clear fashion.
Point 2: Author must show previous reported molecular biology results with validated mechanisms in this manuscript not just graphical mechanism images.
Response 2:
Thank you for your suggestion, we understand that the figures represent more than just one paragraph section in the manuscript. Your comment on validated mechanisms are mentioned in different paragraphs throughout the manuscript. Based on your suggestion we added the relevant references in the figures, to reference the described mechanisms. Line 352, 353, 650
Point 3: Author must describe in Figure 2 caption, whether this is their unpublished result or taken form somewhere else. If it is taken from other published source, please mention copyright detail in the caption. It must be same for all relevant figures.
Response 3:
Thank you for this observation. In the caption of figures 2, 3, 5 added the reference of these images, in line 466, 494, 782, respectively.
Point 4: Experimental data related to immunomodulation must be added with copyright information in this manuscript.
Response 4:
Thank you for your comment. We added relevant references to figure 3, which were referenced in the immunomodulatory paragraph. Line 352
We added relevant references to figure 4, which were referenced in the immunomodulatory paragraph. Line 650
Point 5: At the end section ‘Future Prospective and Conclusion’ must be included in this manuscript.
Response 5:
Thank you for your correct comment. Apologies for our omission. This has been added, starting in line 1120.
Round 2
Reviewer 1 Report
The authors addressed all my comments, thanks!
Just some notes for the eventual proofs.
1) current line 638, the [REF] is still missing.
2) new line 815 (Walter-]. Ref.??
3) NETosis in figure 4 instead of netosis.